# Silver Ions as a Tool for Understanding Different Aspects of Copper Metabolism

**DOI:** 10.3390/nu11061364

**Published:** 2019-06-17

**Authors:** Ludmila V. Puchkova, Massimo Broggini, Elena V. Polishchuk, Ekaterina Y. Ilyechova, Roman S. Polishchuk

**Affiliations:** 1Laboratory of Trace elements metabolism, ITMO University, Kronverksky av., 49, St.-Petersburg 197101, Russia; massimo.broggini@marionegri.it (M.B.); epolish@tigem.it (E.V.P.); ikaterina2705@yandex.ru (E.Y.I.); 2Department of Molecular Genetics, Research Institute of Experimental Medicine, Acad. Pavlov str., 12, St.-Petersburg 197376, Russia; 3Department of Biophysics, Peter the Great St. Petersburg Polytechnic University, Politekhnicheskaya str., 29, St.-Petersburg 195251, Russia; 4Laboratory of molecular pharmacology, Istituto di Ricerche Farmacologiche “Mario Negri” IRCCS, Via La Masa, 19, Milan 20156, Italy; 5Telethon Institute of Genetics and Medicine, Via Campi Flegrei 34, Pozzuoli (NA) 80078, Italy; polish@tigem.it

**Keywords:** copper metabolic system, copper/silver transport, silver nanoparticles

## Abstract

In humans, copper is an important micronutrient because it is a cofactor of ubiquitous and brain-specific cuproenzymes, as well as a secondary messenger. Failure of the mechanisms supporting copper balance leads to the development of neurodegenerative, oncological, and other severe disorders, whose treatment requires a detailed understanding of copper metabolism. In the body, bioavailable copper exists in two stable oxidation states, Cu(I) and Cu(II), both of which are highly toxic. The toxicity of copper ions is usually overcome by coordinating them with a wide range of ligands. These include the active cuproenzyme centers, copper-binding protein motifs to ensure the safe delivery of copper to its physiological location, and participants in the Cu(I) ↔ Cu(II) redox cycle, in which cellular copper is stored. The use of modern experimental approaches has allowed the overall picture of copper turnover in the cells and the organism to be clarified. However, many aspects of this process remain poorly understood. Some of them can be found out using abiogenic silver ions (Ag(I)), which are isoelectronic to Cu(I). This review covers the physicochemical principles of the ability of Ag(I) to substitute for copper ions in transport proteins and cuproenzyme active sites, the effectiveness of using Ag(I) to study copper routes in the cells and the body, and the limitations associated with Ag(I) remaining stable in only one oxidation state. The use of Ag(I) to restrict copper transport to tumors and the consequences of large-scale use of silver nanoparticles for human health are also discussed.

## 1. Introduction

Copper is an essential micronutrient that belongs to the group of ubiquitous trace elements [1]. In biosphere, copper has been documented as the third most abundant trace element after iron and zinc. A normal human body (~70 kg) contains about 100 mg of copper, 10 times less than the amounts of iron (4–5 g) and zinc (1.4–2.3 g) [2]. However, the biological role of copper in aerobic organisms cannot be underestimated. The ground state electron configuration of the copper atom is [Ar]3*d*^10^4*s*^1^. Similarly, to other group 11 elements (Ag, Au), only one electron is left in the 4*s* shell, allowing the 3*d* shell to close (3*d*^10^) and producing a more stable configuration. This explains, to a large extent, the transient properties of copper. Copper has two stable oxidation states, Cu(I) ↔ Cu(II), which are reversible under physiological conditions. The redox potential of this couple is widely used for the catalysis of redox reactions, which involves molecular oxygen [3,4,5,6] and one electron transfer [7]. Consequently, in the biosphere, global energy production (respiration and photosynthesis) is not feasible without copper. In mammals, copper operates as a structural and catalytic cofactor of enzymes (cuproenzymes) involved in vitally important processes, including protection from active oxygen metabolites, oxidative phosphorylation, connective tissue biogenesis, post-translational neuropeptide activation, neurotransmitter synthesis, and transmembrane iron transport [8,9]. In addition, copper regulates angiogenesis [10], the number of intracellular signaling pathways [11,12,13,14], mitochondria-mediated apoptosis [15], and communication between neurons and astrocytes [13,16,17] as well as participating in the regulation of transcription [18]. Therefore, some functions of copper resemble secondary messenger functions. It has been shown that mammalian odorant receptors in olfactory sensory neurons responsible for recognizing strong-smelling sex attractants [19] and thiol compounds [20] have sites that chelate Cu(I), the loss of which leads to a loss of receptor activity. This phenomenon can be attributed to copper’s role as a co-receptor or biosensor. 

Since copper carries out its essential functions during changes in oxidation state, it is a potential source of electrons for the catalysis of Fenton reactions. The products of such reactions induce oxidative stress, in turn, damaging the cells, for example, in ionizing irradiation [21]. In the active centers of enzymes, copper is coordinated by many various ligands (as many as six) and is strongly held in both oxidation states [22]. Copper mobilization mechanisms evolved in parallel with a safe intracellular copper transport system (CTS). The CTS is highly conserved among different species and comprises transmembrane and soluble Cu-transporting and Cu-reserving proteins and peptides. CTS members contain Cu-binding motifs that coordinate Cu(I) with the help of a few sulfur atoms in cysteine and methionine. Usually, the number of Cu/S coordinating in the Cu-binding protein domains is two, but this can vary from one to four. The transfer of copper between such domains happens in the direction of increased affinity and without valence change, which requires the activity of the metallothionein/glutathione redox cycle [23]. Components of the CTS not only deliver copper to the cuproenzymes but also facilitate its integration into the active sites [24], exchange copper between themselves [25], generate local copper concentration gradients [26], control copper functions via its redistribution between organelles/compartments, and regulate its recycling and excretion [27,28]. According to this, the dynamic behavior of the copper in the CTS can be considered as an additional function that might be tightly related to intracellular signaling [29]. 

Currently, aberrations in the system supporting the homeostasis of copper are considered among the reasons for the development of neurodegenerative, oncological, and cardiovascular diseases [30,31,32,33]. This quite numerous and heterogeneous group of diseases can be defined as copper-related disorders (CRD). Only some CRDs are caused by mutations in genes that encode proteins with well-established functions in the CTS. These include Menkes [34] and Wilson [35] diseases, occipital horn syndrome, Menkes ATPase-related distal motor neuropathy [36], MEDNIK syndrome [37,38], aceruloplasminemia [39], and amyotrophic lateral sclerosis [40]. The contributions of Cu-dependent mechanisms have been documented widespread pathologies, such as Alzheimer’s [41] and Parkinson’s [42] diseases, diabetes mellitus [43], cancer [44], and cardio-vascular disorders [31]). However, these mechanisms are yet to be fully understood [45]. 

In this context, the experimental use of silver ions might help to better characterize the Cu-dependent processes behind the pathogenesis of these disorders. The silver atom has a ground state electron configuration of [Kr]4*d*^10^5*s*^1^. Again, one of the electrons of the top 5*s*-shell is borrowed into the 4*d*-shell, producing an energetically favorable closed 4*d*^10^ shell. Thus, the structures of the valence shell of silver atom and its respective silver ion (Ag(I)), which is formed by the loss of the single top *s*-shell electron, are highly like those of copper and the Cu(I) ion. Given the close values of ionic radii, the coordination properties of Ag(I) are similar to those of Cu(I) [46,47]. In proteins, the redox-inactive Ag(I) may occupy only Cu(I) coordination sites. In contrast to copper, silver does not reach the Ag(II) oxidation state in the aquatic environment, and the known individual Ag(II) complexes in the presence of water are instantly restored to Ag(I) [48]. As an antibacterial agent, silver has long been used in medical practice. In recent years, the production of silver nanoparticles (AgNPs) has increased exponentially, and they are used not only in engineering but also in various fields of biomedicine, including acting as substitutes for antibiotics [49]. This has led to increased silver contents in both the environment and the human body itself, contributing to ecotoxicity, primarily due to the production of reactive oxygen species (ROS) [50,51]. Recent research suggests that silver toxicity may be the result of Cu(I) and Ag(I) forming non-identical coordination spheres in CTS proteins, causing the integration of Ag(I) into the Cu(I) metabolic system to result in copper dyshomeostasis [46,47]. Studying the influence of silver ions on the copper metabolic system should help in assessing the undesirable impact of silver use on the biosphere and human health and will allow yet unknown mechanisms of copper homeostasis to be identified. This review focuses on the prospects of silver as a tool for investigating in new aspects of copper metabolism and on the adverse consequences of silver interference with copper transport, distribution, and turnover in mammals.

## 2. Expedients Used to Treat Biological Objects with Silver Ions

For more than 50 years, silver ions have been used to study the systems that control the copper turnover and the mechanisms supporting its homeostasis [52]. For this purpose, silver nitrate is used in the form of a low-toxicity, highly soluble silver salt (Ksp = 1.44 at 25 °C). Silver nitrate solutions have been added to cell culture medium, injected into the tail vein of rat [53], intraperitoneally [54], subcutaneously [55], directly into the stomach, or into a 7.0 cm segment of intestine immediately distal to the pylorus [52], and added to fodder [56]. However, in many studies it has not been considered that Ag(I) from AgNO_3_ in the cell culture medium, food or the body extracellular spaces, is immediately converted to poorly soluble AgCl (Ksp = 1.78 × 10^−10^ at 25 °C) while not compensating the possible toxic effect of the nitrate ion. To avoid undesirable effects, the silver chloride grains are added to powdered moistened fodder [57] or the cell growth medium is saturated by AgCl and then diluted with medium [58]. The concentrations and total doses of silver used in studies vary over a wide range (0.2–50 mg/kg body weight daily, from a single dose to a chronic keep on Ag-diet). In addition to inorganic silver compounds, silver acetate and coordinated silver in N-heterocyclic carbene complexes [59] are used. Silver ions from all the listed compounds are picked up by the CTS. 

## 3. Silver Transport through Extracellular Pathways

The results of the pioneering in vivo studies showed that even though in the gastrointestinal tract (GIT) silver ions should form poorly soluble silver chloride, Ag(I) enters the body. It is likely that Ag(I) is successfully absorbed by enterocytes through coordination by amino acids, short peptides, and possibly bacterial chalkophores produced by symbionts in the GIT [60]. Pulse-chase experiments revealed that the silver is first delivered to the liver, then it was found in peripheral blood in the protein fraction, and only later it was detected in other organs [61]. It was noted that silver is selectively distributed among the organs. It mainly accumulates in the liver and poorly penetrates beyond the cell barriers [62]. Free silver ions, or ions associated with low-molecular substances, were not detected in the blood serum. In Ag-treated mouse liver, silver was found to be associated with both metallothionein and high-molecular-weight proteins residing in the membranes of the secretory pathway [63]. Ag treatment leads to a reduction of two parameters related to copper status in the serum—the total copper concentration and oxidase activity associated with ceruloplasmin (Cp)—but it does not affect the Cp protein concentration [55,64,65]. In rats that received Ag fodder over a long period of time, silver appeared in the urine [63,65], as is observed for copper in Wilson disease [66]. This indicates a substantial overlap between the pathways and molecular players that distribute copper throughout the body. 

In this context, silver was employed to investigate the Cp properties related to its actions as a copper carrier. Fairly old studies revealed that constant feeding of female rats with Ag-rich food during pregnancy led to the loss of Cp activity and caused developmental abnormalities or prenatal death of embryos or 100% mortality of the pups within the first 24 h of life [57]. On the other hand, injections of human holo-Cp into pregnant rats strongly attenuated Ag-mediated embryotoxicity [57]. These findings indicate that Cp operates as a copper carrier and supplies copper to extrahepatic cells. The issue of whether Cp is indeed an extracellular copper transporter has been discussed for several decades. Several lines of evidence suggest that Cp has a copper-transporting function. First, it has been shown that all copper that is adsorbed in the GIT enters the liver and then returns to the bloodstream within Cp [67]. Second, injected [^3^H]Cp was detected in different organs of copper-deficient rats that had scarce levels of their own Cp [68]. Third, Cp can transfer copper ions into cultured cells [69]. Finally, molecular dynamics predicts a specific interaction between the high-affinity copper transporter 1 (CTR1) ectodomain and Cp sites that connect liable copper ions [70]. 

The main objection against the copper-transporting function of Cp comes from the observation that extrahepatic cells do not manifest significant copper deficiency in patients with aceruloplasminemia, an autosomal recessive hereditary disease that develops due to mutations in the *Cp* gene [39]. However, evidence for this objection is not very strong, because mammals accumulate copper in the liver during the embryonic and early postnatal period to distribute it to the organs, and further maintenance of copper might be supported by its recycling. Therefore, it is difficult to create exogenous copper deficiency in adult mammals [71]. This also explains why, during aceruloplasminemia, the main pathologic manifestations are caused by the loss of ferroxidase functions of Cp rather than by the loss of the copper-transporting function of Cp [72]. However, despite this, the copper-transporting function of Cp appears to be critical for newly forming and rapidly growing cellular communities (like embryos or tumors). It may be possible for Ag(I) to be used to study some aspects of aceruloplasminemia related to ferroxidase activity and the copper-transporting function of Cp during different periods of ontogenesis.

Moreover, it is worth noting that in lactating rats, the silver radioactive isotope [^110^Ag], enters the mammary gland cells and into the hepatocytes with kinetic characteristics similar to those of [^64^Cu] [61,73]. Ag(I) included in Cp will disturb its oxidase and ferroxidase activities [65]. Milk Ag-Cp might compromise the copper metabolism of newborn pups, thus helping to highlight yet unknown details of copper transport and turnover in post-natal development [74]. These data suggest that silver could be used as a powerful tool to investigate copper metabolism in newborns. 

## 4. Pathways of Silver Import through the Plasma Membranes

### 4.1. CT R1

Copper uptake from the extracellular space mainly relies on the plasma membrane protein CTR1 (Figure 1) [75,76]. CTR1 operates as a key component of the safe transport system of copper in all eukaryotes and is ideally adapted for the transport of silver ions.

The physiologically active form of CTR1 is a homotrimer [77,78,79,80], and the CTR1 monomer is a type I transmembrane protein. The extracellular N-terminal portion of mammalian CTR1 contains three copper-binding motifs. Motifs 1 and 2 are divided by N-glycosylation sites, while the polyglycine linker is situated between motifs 2 and 3. Motif 1 is formed by Met and His, motif 2 consists of His residues, and motif 3 contains (Met)*n*-X-Met clusters, where *n* can vary from 1 to 6. Only motif 3 appears to be both essential and sufficient to complement the loss of free copper ion transport in yeasts [81]. According to the Pearson chemical hardness principle [82], copper-binding motifs 1 and 2 of CTR1 might be involved in Cu(II) binding from extracellular donors. Cu(I) and Ag(I) exhibit high affinity to copper-binding motifs 1 and 3 of CTR1 [83]. The ability of motif 3 to form selective binding sites with Cu(I) and Ag(I), but not with bivalent metals, has been confirmed experimentally [84]. The CTR1 monomer contains three α-helixes, which are highly conserved in all eukaryotes and form three transmembrane domains (TM1, TM2, and TM3). In the homotrimer, nine α-helices of identical subunits form a cuprophilic pore, which aligns with the threefold central symmetry axis [77,85]. At the extracellular side of the pore, three pairs of conserved methionine residues in the three TM2s form two thioether rings separated by one turn of the α-helix. These serve as highly selective filters for Cu(I) ions. Each ring creates a coordinate sphere of three sulfur atoms, which can coordinate two Cu(I) or two Ag(I) complexes [48]. The copper/silver ions captured in the thioether trap move inside the pore through an electrostatic gradient in ATP-independent manner [85]. 

The last three amino acids of the short CTR1 cytosolic domain form a His-Cys-His stretch, which contains a sphere of two nitrogen atoms and a sulfur atom for the coordination of Cu(I)/Ag(I) [48]. Thus, the parallel use of Cu(II)/Cu(I) and redox inactive Ag(I) shows that CTR1 only imports Cu(I)/Ag(I), while Cu(II) remains bound to the CTR1 ectodomain and has to be oxidized for import through CTR1. 

In mammals, *CTR1* gene is expressed in all cells, and the CTR1 protein serves as the main importer of copper from the bloodstream [86]. However, the pathways of copper absorption in the GIT remain unclear. The first candidate for participation in this process is CTR1, which localizes at the apical membrane of enterocytes. Intestinal epithelial cell-specific CTR1 knockout affects copper accumulation in peripheral tissues and causes hepatic iron overload, cardiac hypertrophy, and severe growth and viability defects [87]. Moreover, a previous study showed that mice fed a copper-deficient diet had elevated levels of apical membrane CTR1 protein [88]. Another study showed that although *CTR1* is expressed in enterocytes, the CTR1 protein resides at the basolateral membrane and [^64^Cu] is taken up through this surface domain of enterocytes. Thus, basolateral CTR1 has been proposed to participate in the delivery of copper/silver from the blood to the intracellular proteins of enterocytes [89]. 

### 4.2. CTR2

CTR2 (low-affinity copper transporter 2) is another potential carrier of copper and silver through the plasma membrane. Its gene was identified by a structural similarity with the *CTR1* gene [75] and presumably, *CTR2* gene arose as a result of duplication and subsequent functional divergence [90]. Further, *CTR1* and *CTR2* are situated in the same chromosome and DNA strand. CTR2 stimulates copper uptake, is expressed in the cells of different internal organs, in the brain, and in the placenta [91] and is localized to late endosomes and lysosomes [92,93] as well as the plasma membrane [91]. The amino acid composition, secondary structure, and topology of CTR2 and CTR1 monomers, as well as their ability to form homotrimers and cuprophilic pores, are highly identical [90]. However, unlike CTR1, CTR2 lacks the ectodomain with copper-binding motifs and hence cannot bind to Cu(II) in the extracellular space. However, the cuprophilic pore of CTR2 still contains two thioether rings composed of two methionine residues, allowing the transport of Cu(I) and Ag(I) across the cell membrane [91]. This might explain why copper and silver absorption decreases in cells lacking *CTR1* and divalent metal transporter 1 (*DMT1*) but is not suppressed completely [94,95]. 

In addition to the transfer of copper through the plasma membrane, several other functions of CTR2 have been revealed including the regulation of copper influx via the induction of CTR1 ectodomain cleavage [96] and participation in copper mobilization from endolysosomal organelles to the cytosol [93]. CTR2 also plays a role in limiting cisplatin accumulation [97], which is in line with a hypothesis predicting that the transfer of cisplatin through cuprophilic pore of CTR1 requires the binding of copper ions with the ectodomain [98]. 

### 4.3. DMT1

The list of copper importers also includes divalent metal transporter 1 (DMT1), a member of the proton-coupled metal ion transporter family, which mediates the transport of ferrous iron from the lumen of the intestine into the enterocytes. DMT1 consists of the only subunit with 12 α-helices, which form transmembrane domains. Both the N- and C-termini of DMT1 are oriented toward the cytosol [99]. The role of DMT1 in importing copper is supported by data showing that knockout of *CTR1* stimulates the expression of *DMT1* and vice versa [94,95]. DMT1 is mainly localized at the apical surface of the enterocytes and the plasma membranes of cells from other organs [100,101]. It plays a relevant role in physiological Cu(I)/Cu(II) entry. However, silver does not inhibit DMT1-mediated copper uptake [102]. Therefore, the participation of DMT1 in the transfer of silver into the cells seems unlikely.

### 4.4. Other Transporters

In parallel with recognizing the roles of CTR1, CTR2, and DMT1 in transporting copper from the GIT and the blood circulation, a growing body of evidence suggests the existence of an alternative pathway of copper absorption, which appears to be independent from the above-listed transporters. Initially, Lee and coworkers demonstrated that CTR1 knockout cells from mouse embryos remained capable of importing copper [103]. The CTR1-independent copper transport was shown to be saturable, time-, temperature-, and pH-dependent, ATP-independent, and inhibited by biological Cu(II) ligands. Moreover, Ag(I), which is transported to the cells via CTR1 [104], did not inhibit the copper import through the CTR1-independent pathway. Thus, the authors concluded that the CTR1-independent copper transport pathway preferentially transports Cu(II) over Cu(I) [105]. Later, enterocyte- and fibroblast-like cells with CTR1 deletion were incubated with the specific DMT1 inhibitor to show that Cu(II) and Cu(I) uptake still occurred and, importantly, this Cu(I) uptake was not inhibited by Ag(I) [105]. Thus, silver can apparently flow into the cells through CTR1 and CTR2, but not through other copper importers.

## 5. Interplay between Silver and Pathways Driving Intracellular Copper Distribution 

Cu(I), transferred through the CTR1 pore, is bound by the cytosolic His-Cys-His motif [106,107], which is involved in both copper coordination and the transfer mechanism to cytosolic Cu(I) chaperones. The transfer of copper from the cytosolic domain of CTR1 to apo-chaperones occurs on the *cis*-side of the plasma membrane through direct protein–protein contact. Holo-chaperones then transfer the copper to the places where it is loaded into cuproenzymes including the mitochondria, secretory pathway compartments, and cytosolic sites where superoxide dismutase 1 (SOD1), a key enzyme in the antioxidant system of aerobic organisms, resides (Figure 1). The list of Cu(I)-chaperones comprises several well-characterized members. For example, antioxidant protein 1 (in humans, ATOX1 or HAH1) ferries copper to the Cu-transporting ATPases, ATP7A, and ATP7B, which transfer copper to the luminal trans-Golgi spaces for the metalation of secretory cuproenzymes. CCS (copper chaperone for Cu/Zn-SOD1) delivers copper to the active catalytic centers of SOD1. Cox17 transfers copper to the mitochondria, where it is required for the formation of mature cytochrome-*c*-oxidase (COX) and SOD1 localized in the mitochondrial intermembrane space (IMS).

It is worth noting that Cu-chaperones can also obtain Cu(I)/Ag(I) from CTR2, despite it lacking the C-terminal His-Cys-His motif. To do this, the chaperones use lipophilic sites on their surface, which allow them to bind the cell membrane and receive Cu(I)/Ag(I) ions at the exit from the CTR2 pore. Structural, genetic, and biochemical data on Cu-chaperones have been analyzed in several recent reviews [24,25,26,108,109,110,111]. Here, we will discuss the findings related to the role of Cu-chaperones in silver transport through mammalian bodies.

### 5.1. ATOX1 

ATOX1, a small cytosolic protein, contains 68 amino acid residues folded into a βαββαβ-plait with a single Cu-binding Met-Xaa-Xaa-Cys-Xaa-Xaa-Cys motif coordinated with Cu(I) on a surface-exposed loop. Holo-ATOX1 is more compact than apo-ATOX1, and it has two different conformations through which it can fulfill its dual roles in copper binding and transfer [112,113,114]. In ATOX1, Ag(I) binds in diagonal coordination to the two cysteine residues of the Cu(I) binding loop and shows high affinity for this protein. X-ray absorption spectroscopy has shown that in the ATOX1 homodimer, the geometric characteristics of the bonds in the coordination sphere differ from those in the [Cu(I)(Atox1)_2_] complex [115,116]. Several lines of evidence suggest that ATOX1 efficiently participates in the transfer of both Cu(I) and Ag(I) between different chains of CTS. Apo-ATOX1 is capable of binding to Cu(I)/Ag(I) through coordination with histidine and cysteine residues in the cytosolic domain of CTR1 [114]. ATOX1 belongs to the group of so-called moonlighting proteins [117]. The main ATOX1 function consists of delivering copper to ATP7A and ATP7B, which then transport copper ions across the membranes of the biosynthetic pathway (Figure 1). Moreover, ATOX1 has been shown to exchange Cu(I)/Ag(I) ions with CCS and receive Cu(I)/Ag(I) from metallothioneins [25]. Besides the role in delivering copper to the secretory pathway of the cell, holo-ATOX1 seems to be capable of transporting copper to the nucleus with the help of the p53 protein [118]. ATOX1 has also been suggested to act as a copper-dependent transcription activator for the *SOD3* gene. Indeed, Itoh and colleagues demonstrated that ATOX1 is bound to the *SOD3* promoter in a copper-dependent manner in vitro and in vivo [119]. Apo-ATOX1 can extract Cu(I) from the ATP7B metal-binding motif and downregulate its activity [28]. It plays an essential role in the copper export pathway, and it is possible that the ratio of apo- and holo-forms of ATOX1 is involved in the coupling of redox homeostasis to intracellular copper distribution [28]. ATOX1 participates in the differentiation of neurons through local changes in copper concentration [120] and was also recently shown to promote cell migration in breast cancer [121]. Therefore, intracellular transport pathways of Ag(I) can be mediated by the substitution of Cu(I) in ATOX1 and further delivery of ATOX1-bound silver to different intracellular compartments.

### 5.2. Copper Delivery to the Cellular Secretory Pathway

In mammals, the transfer of copper from the cytosol to the secretory pathway or extracellular space is carried out by two Cu-transporting P1-type ATPases, ATP7A and ATP7B, which normally reside in the TGN (trans-Golgi network) compartment (Figure 1). ATP7A and ATP7B have also been called Menkes ATPase and Wilson ATPase, respectively, due to the inherited diseases that are caused by mutations in the genes encoding these proteins [122,123,124,125]. ATP7A and ATP7B proteins are very similar in terms of primary structure and domain topology. Therefore, their functions, catalytic cycles, and mechanisms of copper transfer through the membrane are also highly similar [126]. ATOX1 serves as a cytosolic donor of Cu(I) and Ag(I) for both ATP7A/B. There is a lack of strong specificity between the luminal sites of Cu(I)-ATPases that transmit copper and sites of apo-enzymes that receive copper in the secretory pathway. 

*ATP7A* gene is expressed in all organs, including the newborn liver. The ATP7A protein normally resides in the TGN, where it loads copper onto newly synthesized cuproenzyme that is moving through the secretory pathway. In response to an increase in copper concentration, ATP7A moves toward the plasma membrane to promote the excretion of excess copper from the cell [127]. The Cu-transporting activity of ATP7A is required for the delivery of dietary copper from the enterocytes to the blood [128] and has been shown to participate in copper transfer from astrocytes to neurons [129]. The cuproenzymes, which obtain copper from ATP7A localized in extracellular spaces (blood, extracellular matrix, cerebrospinal fluid, and vesicles derived from the Golgi complex), or are inserted into the membrane. The enzymes to which ATP7A transfers copper belong to several subclasses (Table 1) and have different active center structures. The His/Met-rich segment of the first ATP7A extracytosolic loop, which binds Cu(I) and Ag(I), is likely to play a key role in the metalation of cuproenzymes [130]. Interestingly, a fragment of the second extracellular loop specifically binds to the Cp (*K_d_* = 1.5 × 10^−6^ M) and, according to protein footprinting, protects a fragment of the Cp domain 6 [131].

Notably, these cuproenzymes are likely to have different affinities for silver (Table 2), which might be employed to selectively inhibit their catalytic activity and hence, to study their functions in the corresponding biological context.

The expression and physiological functions of ATP7B are mainly related to its role in the liver, where it drives the sequestration of excess copper and its excretion through bile [132,133]. Moreover, ATP7B contributes to the maintenance of copper levels in the blood through the synthesis and secretion of Cp [134]. Finally, some reports suggest that ATP7B might be involved in the synthesis of coagulation factors VIII and V [135,136]. Besides the liver, ATP7B expression has been detected in cells of neural origin and vascular endothelial cells [137,138]. It has been assumed that segments of the luminal loops of ATP7B participate in the direct transfer of copper to the active sites of intact Cp [139,140].

Cp is the main protein metalized by ATP7B. However, in cells with ATP7B knockout, ATP7A efficiently substitutes for ATP7B in loading copper onto the newly synthesized Cp [141]. In addition, ATP7A loads copper onto Cp in the liver during early postnatal development when ATP7B is poorly expressed in the hepatocytes [142]. *Cp* gene encodes two mRNA splice variants, which are translated into secretory Cp and membrane-bound Cp with a glycosylphosphatidylinositol anchor (GPI-Cp) [143]. The expression of *ATP7A* and *GPI-Cp* coincide in different brain regions [137] and in the mammary glands [144], suggesting that GPI-Cp is likely to be mainly metalized by ATP7A rather than ATP7B. In neuronal cells, both *ATP7A* and *ATP7B* can be expressed simultaneously and supply copper to dopamine-beta-hydroxylase in a selective manner, which depends on its localization [145]. Despite structural similarities and co-participation in maintaining the homeostasis of copper, the loss of ATP7B function does not result in an elevation of ATP7A mRNA level [146] and vice versa [147]. 

N-terminal Cu-binding domains of both ATP7A and ATP7B contain core -Cys-Xaa-Xaa-Cys- stretches, which bind Cu(I) and Ag(I) in a similar manner [148]. These domains receive silver ions from Ag-ATOX1 and participate in their transfer to the lumen of the Golgi compartment or their excretion through the bile (our unpublished results). The silver ions that are transferred to the secretory pathway can be incorporated into the cuproenzymes synthesized de novo.

Indeed, it has been shown that silver ions were included in the Cp molecules synthesized in the liver [56,63,65]. Cp is a blood serum N-glycoprotein, which consists of a single polypeptide chain with a molecular weight of 132 kDa containing 1046 amino acid residues (in human) [134]. About 95% of extracellular copper has been reported to associate with Cp [149]. Cp belongs to the family of multi-copper blue ferroxidases [5,150]. Its active centers contain six copper ions, of which amino acids of domains 3, 4, and 5 form mononuclear centers, and the amino acid residues of domains 1 and 6 form three-nuclear centers [150,151]. Cp belongs to the category of moonlighting proteins [152,153]. Its major function is to facilitate iron redox transitions, which are required for transferrin receptor- and ferroportin-mediated transport of iron through the membranes [154]. In vivo, Cp oxidizes dopamine, serotonin, epinephrine, and norepinephrine, thus inactivating them [134,152]. Cp is an acute-phase protein, as its level increases by several times during processes such as inflammation, ovulation, pregnancy, and lactation [155]. Cp also demonstrates weak antioxidant activity toward ROS and regulates the oxidative status of neutrophils [156].

Cp efficiently binds to silver, which affects its catalytic activity. It was shown that Cp in blood serum from Ag-fed rats exhibits low oxidase and ferroxidase activity. In addition, inactive Cp has been found to contain one to three silver ions per molecule as molten globule [63,65,157]. Presumably, the substitution of copper with silver in the active sites of Cp happens due to the presence of three cysteine residues, each of which creates a coordination area for Cu(I)/Ag(I) with two histidine residues (Table 2). Despite the extensive investigation of active Cp sites using different approaches, including biochemical, chemical, biophysical, and molecular dynamics, some aspects of its enzyme activity and participation in various processes remain unsolved [158]. Thus, further study is required to determine whether the maturation of Cp in the Golgi requires a strict cooperative order of filling of active centers by copper ions. The use of silver may help to solve this issue.

Despite being catalytically inactive, Ag-bound Cp does not undergo rapid degradation like apo-Cp, which is not loaded with copper. In a previous study, Ag-fed rats did not exhibit a significant decrease in overall Cp levels in either blood or isolated Golgi membranes, while blood copper values and Cp oxidase activity remained barely detectable [56,63,65]. These findings differ strikingly from those related to Cp deficiency in Wilson disease, where the loss of ATP7B function does not allow Cp to be loaded with copper in the Golgi. As a result, apo-Cp is rapidly degraded in ATP7B-deficient animals and patients. Indeed, ATP7B-deficient LEC (Long Evans Cinnamon) rats manifest low Cp levels and activity, while Ag treatment does not further affect Cp abundance and function due to a lack of ATP7B-mediated transfer of Ag(I) to Cp. As in the case of copper, previous research found that Ag in LEC rats was bound to metallothionein, excluded from Cp, and not excreted in bile [63]. Interestingly, this suggests that silver allows Cp oxidase activity to be selectively inhibited without significantly impacting the overall protein expression. This finding should aid in the understanding of the pathologic mechanisms associated with the loss or aberrant modulation of Cp function in different diseases.

A recent study revealed that the consumption of a diet including Ag from the first day of life did not dramatically reduce Cp oxidase activity in the blood. In these animals, Cp activity was about half that in the control group [159]. In vivo pulse-chase experiments revealed that de novo synthesis of [^14^C]Cp in Ag-fed animals occurred even when the liver was isolated from the bloodstream [65]. It turned out that this Cp was synthesized and excreted by the cells of subcutaneous adipose tissue, to which silver was not delivered [160]. Thus, the silver helped to uncover the interorgan control mechanism that supports copper balance in the blood and compensates for the deficit of oxidase Cp.

A growing body of evidence suggests that silver treatment could be of value for various medical purposes. For example, silver might modulate the efficiency of cisplatin chemotherapy, which is widely used to treat solid tumors. As in the case of copper and silver ions, cisplatin uptake into the cells requires CTR1. Thus, modulation of copper status (also with silver) has been considered as an option for the acceleration of cisplatin influx into the cells [161,162,163]. Indeed, it was recently shown that an Ag diet can be successfully used for this purpose [98]. One of the general signs of tumor development, regardless of its nature, is increased copper consumption, which is manifested in the activation of copper metabolism genes [164]. In this context, Ag(I) might interfere with cuproenzyme synthesis and angiogenesis, which both require copper to promote tumor growth. In line with this notion, a diet including Ag was shown to inhibit the growth of human tumors engrafted into nude mice [164]. 

Mutations in *ATP7A* cause copper deficiency, which can lead to the development of several disorders such as Menkes disease, occipital horn syndrome, and ATP7A-related distal motor neuropathy [34]. (His)_2_Cu injections have shown high therapeutic efficiency in patients carrying certain *ATP7A* mutations [165]. The responsiveness of a given patient to such therapy could be predicted using a measurement of the kinetics of copper retention in fibroblasts or in amniotic fluid cell cultures. These in vitro kinetics tests usually require the [^64^Cu] radioactive isotope, which has a half-life of ~13 h. Radioactive silver [^110^Ag], which has a half-life of 250 days, is a valuable alternative that has already been used successfully in diagnostic practice [166]. 

### 5.3. CCS

CCS operates as a Cu(I) chaperone that ferries Cu(I) to SOD1, one of the main antioxidant enzymes in the cell (Figure 1). Moreover, CCS controls the folding and hence the stability of SOD1. The CCS molecule is composed of three structural-functional domains. Domain 1, which contains the Cys-Xaa-Xaa-Cys motif as ATOX1, acquires Cu(I) from CTR1 during CCS docking to the plasma membrane. Domain 2 appears to be structurally similar to SOD1 and plays a key role in CCS–SOD1 protein recognition. Domain 3 is a short polypeptide segment that lacks a secondary structure but contains a Cys-Xaa-Cys motif that is essential for SOD1 homodimerization via S−S bond formation between SOD1 subunits [167,168,169]. Thus, CCS participates in all stages of SOD1 post-translational maturation, from metalation of the de novo synthesized polypeptide to the formation of the active enzyme homodimer. In the cells, enzymatically active SOD1 is mainly localized in the cytosol, with a minor fraction (about 5%) in the mitochondrial IMS [170]. Presumably, mitochondrial SOD1 protects the mitochondria from oxidative stress, which might be caused by ROS as a result of electron leakage from the electron transport chain [171,172]. Active cytosolic SOD1 cannot be imported into the mitochondria and vice versa. Both SOD1 and CCS enter the IMS in apo forms through the translocator outer membrane (TOM) and bind to the IMS receptor MIA40 (mitochondrial intermembrane space import and assembly complex), which promotes the formation of disulfide bonds and concomitant protein folding. In the mitochondria, CCS acquires Cu(I) from an unknown Cu(I) transporter. 

Although CCS can bind to Ag(I) ions, it does not appear to transfer them to the active site of SOD1. In a previous study, Ag-fed rats and mice did not exhibit a significant loss of SOD1 activity in the cytosol and the mitochondrial IMS [65]. Considering that mRNA, protein levels, and SOD1 activity remain intact in Ag-fed animals, we can conclude that the exchange of Cu(I)/Ag(I) is blocked during SOD1 monomer metalation. This might occur because the active site of the SOD1 is formed only by histidine residues (Table 2), which cannot coordinate the Ag(I) [48], or because Ag(I) fails to be oxidized to Ag(II) and hence to donate the electron, which is needed at the last stage of active SOD1 formation. In any case, silver has no toxic impact on SOD1. Similarly, secretory (extracellular) SOD3 exhibits quite similar resistance to silver incorporation [65]. Ag-fed rats do not manifest changes in the activity of SOD3, which is synthesized in endothelial cells and metalized by ATP7A [173]. Considering the high homology of SOD1 and SOD3, we can assume that silver does not replace copper because the active site of SOD3 also consists of histidine residues (Table 2). 

### 5.4. COX17

The cytosolic Cu(I) chaperone COX17 has been identified as an essential component in the biogenesis of COX, a terminal complex of the mitochondrial electron transport chain [174]. COX consists of 13–14 different subunits (SU), three of which (SU1, SU2, and SU3) form a catalytic center. In mammals, they are encoded by mitochondrial DNA and integrated into the inner membrane using the OXA (oxidase assembly translocase complex). The assembly of mature COX is a complex process requiring high accuracy, which relies on numerous accessory proteins [111,175]. COX activity requires hemes (a + a_3_) and three copper ions, which are included in the di-copper centers SU1 and SU2 (also known as Cu_A_ and Cu_B_). The assembly of both centers depends on copper, which is delivered by the COX17 from the cytosol (Figure 1) [29]. COX17, a small soluble protein with a molecular mass of approximately 8 kDa, contains two Cu-binding motifs, C-X_9_-C flanked by two neighbor cysteines, which cooperatively bind four Cu(I) ions into a Cu_4_S_6_ complex. Ag(I) apparently cannot be embedded into the COX17 molecule at any stage of holo-COX17 formation [176]. It seems that holo-COX17 ferries copper ions from the cytosol toward the mitochondria. Then, COX17 must be unfolded for TOM40-mediated transfer to the IMS, where it subsequently recovers its appropriate 3D structural organization in a MIA40-dependent manner and binds to four Cu(I) complexes [108,177]. 

Holo-COX17 operates as a copper donor for both COX mitochondrial SUs. Each SU receives copper ions from COX17 through the different systems of mitochondrial Cu-chaperones. In SU1 (Cu_B_ center), this function is executed by COX11 through three invariant residues of the histidine which form the coordination sphere for the copper ion [109,178]. SU2 (Cu_A_ center) receives two copper ions from Cu-chaperones SCO1/2, which, in turn, obtain Cu from COX17. SCO1/2 promote the oxidation of Cu(I) to Cu(II), which is required for the integration of copper ions into the COX active site. In mature COX, two copper ions are connected by the –SH of two Cys residues. As a result, a unique electronic structure is formed that allows them to carry out the oxidation of one electron.

This suggests that Ag(I) cannot be transferred to the active sites of COX because COX17 does not bind silver ions, and coordination spheres in SUs do not possess enough affinity for Ag(I). Indeed, COX activity was shown to be unaffected in rats and mice receiving Ag-fodder [58]. In contrast, in proteo- and eubacteria, Ag(I) were found to suppresses COX activity via direct interaction with membranous copper transporting proteins [179]. At the same time in vivo studies suggest that mammalian hepatic mitochondria are capable of accumulating silver, most of which resides in the mitochondrial matrix [58,62]. Therefore, the delivery of silver ions to the mitochondria apparently occurs through COX17-independent pathways, which are probably also used by copper. Mitochondria have been seen to accumulate copper both under physiological conditions (for example, in livers of newborns) [142] and during the development of Wilson disease [66]. The outer membranes of mitochondria can apparently host DMT1, which transports iron, copper, and manganese ions [99,102,180,181]. It might potentially deliver copper (but not silver) ions to the mitochondria. However, DMT1 knockout does not reduce mitochondrial levels of copper [94]. The transfer of copper through the inner membrane is executed by a phosphate transporter, PIC2, in yeast [182] and by its mammalian ortholog encoded by *SLC25A3* [183]. The assembly of active COX is associated with the expression of this gene. Since Cu_A_ and Cu_B_ are metalized in the IMS while PIC2/SLC25A3 transports copper to the matrix, a copper transporter from the matrix to the IMS is required. Anionic fluorescent molecular complex CuL (copper ligand) has been proposed to play this role, which requires CuL shuttling in the cytosol ↔ IMS ↔ matrix directions [184,185,186]. It is likely that CuL also participates in Ag(I) transfer. Moreover, the use of silver ions might help to reveal yet unidentified mitochondrial transporters of Cu(I)/Ag(I). This, in turn, would contribute to the understanding of other mechanisms used by mitochondria to support the overall homeostasis of copper in the cell.

## 6. Interference of Silver Nanoparticles (AgNPs) in Copper Metabolism of Eukaryotes

The properties of silver, high antibacterial activity, excellent thermal and electrical conductivity, make it a widely used metal, and fabrication of AgNPs is economically beneficial. An uncontrolled increase in the application of AgNPs in various areas (technical industry, textile production, agriculture, food industry) has inevitably led to an increased risk of human contact with them in everyday life. AgNP bioactivity indicates chemical instability as a result of the conversion of Ag(0) to Ag(I). In the environment, AgNP corroding is accompanied by sulfidation and chlorination, with the formation of practically insoluble silver salts [187]. If the transformation of AgNPs were to stop at this stage, the AgNPs and their transformation products would not pose threats to humans and the environment. However, silver ions from AgNPs interfere with cellular metabolism, perhaps due to the presence in the biological media of electron carriers, amino acids, and small peptides capable of coordinating Ag(I) by repeatedly increasing the solubility of silver. This raises issues related to the safety of AgNPs for the environment and human health [188]. Therefore, a growing number of studies are aimed at determining the toxic impact of AgNPs on molecular processes in the cells and their mechanisms. The relationships between the bioactivity of AgNPs and their linear size, surface shape, corrosion rate, aggregation state, stability, and biodegradability have been studied [50]. Most investigations have been performed on prokaryotes as targets of AgNP antibacterial action, and cultured cells of higher eukaryotes as models for assessing the toxicity of AgNPs in mammals.

In recent years, more in vivo studies of the effects of AgNPs on cell and molecular processes in higher eukaryotes have been carried out, predominantly on animal models with a short lifespan, well-studied stages of ontogenesis, sequenced genomes, and inexpensive maintenance (such as *Danio rerio, Caenorhabditis elegans*, and *Drosophila melanogaster*). AgNPs with different physicochemical properties has been shown to result in decreased lifespan, fertility, growth, body size, and locomotion [189,190,191,192]. It is generally recognized that the toxic effect of particles is based on AgNP-mediated oxidative stress [50,51]. AgNPs overcome intestinal barriers and are absorbed by the cells through clathrin-mediated endocytosis, stimulating lipid peroxidation, DNA and protein damage, and the induction of apoptosis [193,194,195,196]. In response to ROS-mediated oxidative stress, genes involved in heat shock, DNA repair, cytosol (glutathione peroxidase), mitochondrial Mn-SOD2, and autophagy are activated, possibly through the p38 MAPK/PMK-1 pathway [195,197,198]. Interestingly, the levels of copper-containing enzymes (tyrosinase and SOD1) are significantly decreased in invertebrate animals following treatment with AgNPs, despite the copper level in tissues remaining unchanged [199]. It was proposed that silver ions dissociated from AgNPs bind with copper transporter proteins and cause copper sequestration, thus creating a condition that resembles copper starvation [199]. In mice treated with AgNPs, Cp oxidase activity in the blood serum was shown to decrease [200]. However, *Cp* expression and the relative contents of Cp protein in the Golgi complex and in the serum did not change [200]. In addition, treatment with AgNPs did not influence liver SOD1 activity or serum alanine aminotransferase and aspartate aminotransferase content, i.e., AgNPs had no apparent toxic effects in mice. Dark-colored inclusions were observed in the abdominal cavities of the mice, but only in those that received the largest dose of AgNPs [200]. A woman who ingested 1 L of colloidal silver solution (34 mg silver) daily for approximately 16 months as an alternative medical practice showed evidence of argyrosis [201]. The patient had a serum silver concentration of about 381 ng/mL, 25-fold higher than the reference level. In the intercellular space of her sweat glands and hair follicular epithelia, brown-black granules containing silver were deposited, but other signs of toxicity were not observed. In total, the data show that the release of large masses of AgNPs into the environment, e.g., during industrial disasters, will lead to severe consequences. However, moderate concentrations, which can typically be achieved by eating foods containing AgNPs, lead to the interference of silver ions from the AgNPs in copper metabolism, affecting the various processes in the cell (Figure 2). 

Thence, the long-term effects of such interventions have not been assessed, and data obtained from studying the Ag(I) routes in the bodies and cells of mammals are required.

## 7. Conclusions

In sum, this analysis of the existing studies highlights the usefulness of silver for investigating various metabolic pathways that require copper as an essential participant. Moreover, silver itself has started to gain interest from different research fields due to its emerging role in bioengineering, medicine, nutrition, and environmental pollution. Thus, we expect that biological studies focusing on silver will expand to reveal new mechanisms and pathways that are involved in its transport, turnover, and metabolism.

## Figures and Tables

**Figure 1 nutrients-11-01364-f001:**
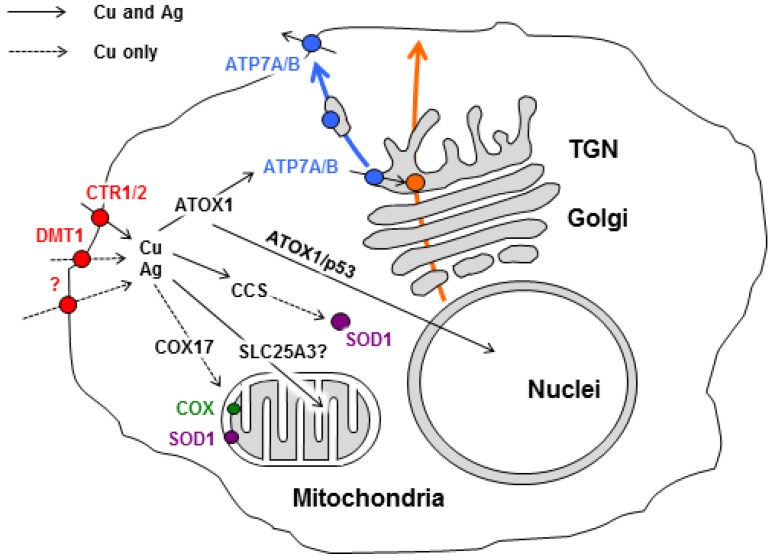
Scheme of copper and silver distribution in a mammalian cell. Copper is taken up via copper transporter 1 (CTR1), divalent metal transporter 1 (DMT1), or the putative transporter (all depicted as red circles). After being imported into the cell, the copper is transferred to chaperone antioxidant protein 1 (ATOX1), copper chaperone (CCS), and cytochrome-*c*-oxidase (COX17), which ferry it (black arrows) to both copper-transporting ATPase (ATP7A/B, blue) in the Golgi, to Cu, Zn-superoxide dismutase (SOD1, magenta) in the cytosol, and to cytochrome-c-oxidase (COX, green) in the mitochondria. Mitochondrial phosphate carrier protein (SLC25A3) transfers copper into the matrix. In the Golgi, ATP7A/B load Cu on newly synthesized cuproenzymes (orange circle), which transport it along the biosynthetic pathway (orange arrow). A significant increase in intracellular Cu induces the export of ATP7A/B (blue arrow) toward the post-Golgi compartments (TGN) and plasma membrane, where it drives the excretion of excessive Cu from the cell. Silver uses similar copper-transporting routes (solid black arrows). However, several copper-transporting pathways cannot be invaded by silver (dashed black arrows).

**Figure 2 nutrients-11-01364-f002:**
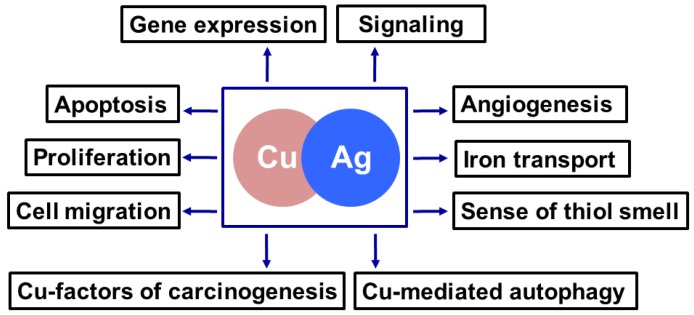
Copper-required cellular processes, in which Cu(I) can be replaced by Ag(I).

**Table 1 nutrients-11-01364-t001:** Catalyzed reactions by cuproenzyme group (source: ExPASy).

Class Name	Catalyzed Reaction	Electrons Transferred to Dioxygen	Cu Atoms Required
Superoxide dismutase 3, EC 1.15.1.1	2 superoxides + 2 H^+^ <=> O_2_ + H_2_O_2_	1 + 1	1
Ferroxidase, EC 1.16.3.1	4 Fe^2+^ + 4 H^+^ + O_2_ <=> 4 Fe^3+^ + 2 H_2_O	4	4 (6)
Peptidylglycine monooxygenase, EC 1.14.17.3	[Peptide]-glycine + 2 ascorbates + O_2_ <=> [peptide]-(2*S*)-2-hydroxyglycine + 2 monodehydroascorbate + H_2_O	2 + 2	2
Dopamine beta-monooxygenase, EC 1.14.17.1	3,4-dihydroxyphenethylamine + 2 ascorbates + O_2_ <=> noradrenaline + 2 monodehydroascorbate + H_2_O	2 + 2	1
Diamine oxidase, EC 1.4.3.22	Histamine + H_2_O + O_2_ <=> (imidazol-4-yl) acetaldehyde + NH_3_ + H_2_O_2_	2	1
Primary-amine oxidase, EC 1.4.3.21	RCH_2_NH_2_ + H_2_O + O_2_ <=> RCHO + NH_3_ + H_2_O_2_	2	1
Protein-lysine 6-oxidase, EC 1.4.3.13	[Protein]-L-lysine + O_2_ + H_2_O <=> [protein]-(*S*)-2-amino-6-oxohexanoate + NH_3_ + H_2_O_2_	2	1
Tyrosinase, EC 1.14.18.1	L-tyrosine + O_2_ <=> dopaquinone + H_2_O 2 L-dopa + O_2_ <=> 2 dopaquinone + 2 H_2_O	4	2

**Table 2 nutrients-11-01364-t002:** Theoretical assessment of the ability of Ag(I) to replace copper in the active centers of the major cuproenzymes of mammals.

Enzyme	Class	Reference Structure(s), PDB ID	Copper Coordination Sphere *	Geometry *	Feasibility of Ag(I) Binding ****
COX	Cytochrome-c-oxidase; EC 1.9.3.1,	5IY5 (cow)	*CuA*; Cu pair, subunit 2, C200 (bridge), C196 (bridge), H161, H204, M207, E198 amide	Distorted tetrahedral for each atom; strong Cu–Cu interaction	Low
			*CuB*; subunit 1, H290, H291, H240, heme	Distorted trigonal pyramidal; Cu–heme interaction	Low
SOD1	Superoxide dismutase, EC 1.15.1.1	1HL5 (human)	H46, H48, H63, H120	Distorted tetrahedral	Low
SOD3	Superoxide dismutase, EC 1.15.1.1	2JLP (human)	H96, H98, H113, H163	Distorted tetrahedral/trigonal	Low
Cp	Ferroxidase, EC 1.16.3.1	1KCW, 2J5W (human)	*Cu21 (blue):* C319, H276, H324	Distorted trigonal planar	Moderate
			*Cu31 ***: H163, H980, H1020 (dioxygen)	Trigonal pyramidal (tetrahedral)	Low
			*Cu32*: H103, H1061, H1022 (dioxygen)	Trigonal (distorted tetrahedral)	Low
			*Cu33*: H101, H978, (dioxygen, water/OH), η_5_-bonding from H103 and H980	Linear (square planar, with η-bonds; tetragonal distorted octahedral)	Low
			*Cu41 (blue):* C680, H637, H685	Distorted trigonal planar	Moderate
			*Cu61 (blue):* C1021, H975, H1026	Distorted trigonal planar	Moderate
			*Cu42 (labile):* H692, D684 (water?)	Angular	Very low
			*Cu62 (labile):* H940, D1025 (water?)	Angular	Very low
Hephaestin (HEPH)	Ferroxidase, EC 1.16.3.1	No data	Putatively similar to Cp, the trinuclear site, Cu21 and Cu41 site are conserved, the presence of blue copper is proven		Moderate for blue sites
Zyklopen (HEPH1)	Ferroxidase, EC 1.16.3.1	No data	Putatively similar to Cp, the trinuclear site and Cu21 site are conserved		Moderate for blue sites
Peptidyl-glycine alpha-amidating monooxygenase	Peptidylglycine monooxygenase, EC 1.14.17.3	1SDW (rat)	*Cu1*, H107, H108, H172	trigonal planar	Low
			*Cu2*, H242, H244, M314 (dioxygen)	Trigonal pyramidal (tetrahedral)	Low
Dopamine beta-monooxygenase	Dopamine beta-monooxygenase, EC 1.14.17.1	4ZEL (human)	H412, H414, M487 (substrate?)	Trigonal pyramidal (tetrahedral?)	Low
Amine oxidase copper-containing 1 (Dopamine oxidase)	Diamine oxidase, EC 1.4.3.22	3HI7	H510, H512; H675, (substrate)	Distorted T-shaped (distorted tetrahedral)	Low
Amine oxidase, copper containing 3 (AOC3)	Primary-amine oxidase, EC 1.4.3.21	2Y73	H520, H522, H684 (substrate, water?)	Distorted T-shaped (seesaw/octahedral?)	Low
Amine oxidase, copper containing 2 (AOC2)	Primary-amine oxidase, 1.4.3.21	No data	Highly similar to AOC3, copper site conserved		Low
LOX	Protein-lysine 6-oxidase, EC 1.4.3.13	1N9E (Pichia pastoris)	H528, H530, H694, modified Y478 (TPQ, *O*-donor)	Distorted tetrahedral	Low to very low
LOXL2	Protein-lysine 6-oxidase, EC 1.4.3.13; putative	5ZE3	H626, H628, H630, Y689 (putative, Zn instead of Cu)	Distorted tetrahedral	Low to very low
LOXL1,3,4	Protein-lysine 6-oxidase, EC 1.4.3.13; putative	No data	Putatively similar to LOX/LOXL2		Low
TYR	Tyrosinase, EC 1.14.18.1	5Z0D, 5Z0F *** (Streptomyces)	*Cu1:* H38, H54, H63, (η_2_-dioxygen)	Distorted trigonal planar (distorted tetrahedral)	Low
			*Cu2:* H190, H194, H216, (η_2_-dioxygen)	Distorted trigonal pyramidal (distorted tetrahedral)	Low
Thiol receptor OR2T11		No data	M115, R119, C238, H241	Distorted tetrahedral	High

* The positions of protein-based electron donor groups are given. Substrate(s) and total effective geometry, which accounts for the substrate, are given in brackets. ** Cu31, Cu32, and Cu33 form a dioxygen binding trinuclear site, provided by eight imidazole groups of histidine residues. During dioxygen binding, the donor groups are preserved, but the coordination geometry changes. *** Only the evolutionary conserved active site is discussed. Other copper ions in these structures are not accounted for. **** Feasibility is based on geometry and coordination spheres. N-donor spheres (His-only) are considered inferior for Ag(I) coordination. Coordination of O-donor ligands (tyrosine, water molecules) and intermetallic bonds (different between metal ions) are also considered as unfavorable for Ag(I) binding.

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
