# Peer review of "Silver Ions as a Tool for Understanding Different Aspects of Copper Metabolism"

_nutrients, 2019, doi:10.3390/nu11061364_

Reviewer 1 Report

In this paper Pushkova et al. review Ag as tool for understanding copper metabolism. The report touches on all the necessary aspects of Cu homeostasis, it is easy to follow and concentrates on the most important aspects.

The reviewer as a few minor concerns that should be addressed before publication of the manuscript.

 In general, this seems to be more like a Cu homeostasis review than a one for Ag. The manuscript should be carefully checked for grammar and spelling errors.

 In the introduction the authors use the words atoms and ions interchangeably. This needs to be corrected (line 67 for example). The chemistry of Cu(I) vs Cu(II) preferring S (soft) over O (hard) could be explained in more detail.

 Line 298. The authors state that the ATOX1-Cu(I) in is 4 coordinate by two ATOX1 molecules. This was however only observed in the crystal structure. The reference they cite shows that 1 ATOX1 molecule coordinates on Cu+ in a diagonal fashion. XAS studies This was first published in 2003 by Ralle, Lutsenko, Blackburn (JBC).

 Line 337. The authors seem to suggest that all enzymes listed in Table1 are enzymes that ATP7A delivers Cu to? This should be corrected; the table rather lists all enzymes that contain Cu as a co-factor.

 Line 442. The suggestion to treat infants or newborns with radioactive Ag is ridiculous!

 In the discussion the authors might want to include a sentence or two about how they think Ag(0) as it appears in the NPs are oxidized to Ag+1.

 Author Response

Dear Reviewer,

we are sincerely grateful to you for comments. They prompted us to critically review some of the conclusions and added more results. All your comments have been considered. Please find our answers in italics below.

 Rev. In general, this seems to be more like a Cu homeostasis review than a one for Ag. The manuscript should be carefully checked for grammar and spelling errors.

Authors. Since we wanted to emphasize the well-known links of copper metabolism, in which silver can interfere, it was necessary to indicate these links. The manuscript was verified by a professional English editor.

Rev. In the introduction the authors use the words atoms and ions interchangeably. This needs to be corrected (line 67 for example). The chemistry of Cu(I) vs Cu(II) preferring S (soft) over O (hard) could be explained in more detail.

Authors. We adhered to the rule: to use the word "atom" when it came to the copper ion, which is in the coordination sphere with an indefinite total charge, and the word "ion", when the copper ion is weakly bound and can easily exchange. We agree that it may introduce confusion, replaced the word “atom” with the word “ion” if it is not about the oxidation state (0). We consider in detail the properties of Cu (I) vs Cu (II) in terms of absolute electronegativity ([83] Parr, R.G.; Pearson, R.G. Absolute hardness: Companion parameter to absolute electronegativity. J. Am. Chem. Soc. 1983, 105, 7512–7516), in our work ([82] Skvortsov, A.N.; Zatulovskiĭ, E.A.; Puchkova, LV. Structure-functional organization of eukaryotic high-affinity copper importerCTR1 determines its ability to transport copper, silver and cisplatin. Mol Biol (Mosk). 2012, 46, 335-347), and refer to it.

Rev. Line 298. The authors state that the ATOX1-Cu(I) in is 4 coordinate by two ATOX1 molecules. This was however only observed in the crystal structure. The reference they cite shows that 1 ATOX1 molecule coordinates on Cu+ in a diagonal fashion. XAS studies This was first published in 2003 by Ralle, Lutsenko, Blackburn (JBC).

Authors. Thank you very much, refinement made.

Rev. Line 337. The authors seem to suggest that all enzymes listed in Table1 are enzymes that ATP7A delivers Cu to? This should be corrected; the table rather lists all enzymes that contain Cu as a co-factor.

Authors. Thank you very much, COX is omitted from table 1.

Rev. Line 442. The suggestion to treat infants or newborns with radioactive Ag is ridiculous!

Authors. The cited paper deals only with studies on cultured fibroblasts, or cells, isolated from amniotic fluid.

Rev. In the discussion the authors might want to include a sentence or two about how they think Ag(0) as it appears in the NPs are oxidized to Ag+1.

Authors. Added.

Reviewer 2 Report

Overall the article is well organized and comprehensive.  It needs to be seriously edited by a native English speaker as there are dozens of grammatical errors throughout.  I feel that a number of statements need additional referencing.  These lines within the text need a reference attached to them: 81, 82, 146, 397, 530, 532, 551, and 553.  Please do not start sentences with "So,". 

Author Response

Dear Reviewer,

we are sincerely grateful to you for comments. The manuscript was edited by the professional English speakers. The additional references were included. Since we made the "Abbreviations" section, the lines shifted. Added references can be seen in new lines.

 1.          Earlier lines 81, 82; lines now 94 – 96; references were ordered.

2.          Earlier line 146; lines now 114; Added: [50] Ackerman, C.M.; Chang, C.J. Copper signaling in the brain and beyond. J. Biol. Chem. 2018, 293, 4628–4635. doi: 10.1074/jbc.R117.000176.

3.          Earlier line 397; line now 402; Added: [149] Giurgea, N.; Constantinescu, M.I.; Stanciu, R.; Suciu, S.; Muresan, A. Ceruloplasmin - acute-phase reactant or endogenous antioxidant? The case of cardiovascular disease. Med. Sci. Monit. 2005, 11, RA48-51.

4.          Earlier line 530; line now 552; Added: [189] Du, J.; Tang, J.; Xu, S.; Ge, J.; Dong, Y.; Li, H.; Jin, M. A review on silver nanoparticles-induced ecotoxicity and the underlying toxicity mechanisms. Regul. Toxicol. Pharmacol. 2018, 98, 231–239. doi:10.1016/j.yrtph.2018.08.003.

5.          Earlier line 532; line now 556; see [50] Akter, M.; Sikder, M.T.; Rahman, M.M.; Ullah, A.K.M.A.; Hossain, K.F.B.; Banik, S.; et al. A systematic review on silver nanoparticles-induced cytotoxicity: Physicochemical properties and perspectives. J. Adv. Res. 2017, 9, 1-16. doi: 10.1016/j.jare.2017.10.008.

6.          Earlier line 551; line now 563; Added: [188] Zhang, W.; Xiao, B.; Fang, T. Chemical transformation of silver nanoparticles in aquatic environments: Mechanism, morphology and toxicity. Chemosphere. 2018, 191, 324-334. doi: 10.1016/j.chemosphere.2017.10.016.

7.          Earlier line 553; line now 568; Added: [199] Chesi, G.; Hegde, R.N.; Iacobacci, S.; Concilli, M.; Parashuraman, S.; Festa, B.P.; Polishchuk, E.V.; et al. Identification of p38 MAPK and JNK as new targets for correction of Wilson disease-causing ATP7B mutants. Hepatology. 2016, 63, 1842-1859. doi: 10.1002/hep.28398.
